# Proposal of Construction Method of Smart Liner to Block and Detect Spreading of Soil Contaminants by Oil Spill

**DOI:** 10.3390/ijerph20020940

**Published:** 2023-01-04

**Authors:** Kicheol Lee, Jungjo Yuu, Jeongjun Park, Gigwon Hong

**Affiliations:** 1Corporate Affiliated Research Institute, UCI Tech, 313 Inha-ro, Michuhol-gu, Incheon 22012, Republic of Korea; 2R&D Center, GoldenPow Co., Ltd., 412 Samseong-ro, Gangnam-gu, Seoul 06185, Republic of Korea; 3Incheon Disaster Prevention Research Center, Incheon National University, 119 Academy-ro, Yeonsu-gu, Incheon 22012, Republic of Korea; 4Department of Civil Engineering, Halla University, 28 Halladae-gil, Wonju-si 26404, Republic of Korea

**Keywords:** soil contaminants, oil spill, smart liner, RSIM, numerical analysis

## Abstract

Soil is an important factor for public health, and when a soil contaminant occurs by oil spill, it has a great impact on the ecosystem, including humans. Accordingly, the area is blocked using a vertical barrier, and various remediation methods are being applied when an oil spill occurs. This study intends to use a smart liner to prevent and detect the spreading of soil contaminants in a situation in which oil spill detection is important. However, the smart liner is in the form of a fiber, so it is impossible to construct it in a general method. Therefore, the roll spreading and inserting method (RSIM) is proposed for smart liner construction. RSIM is a method of installing a supporting pile after excavating the ground and connecting the smart liner vertically to the ground surface. This method is the first method proposed in this study, and the design and concept have not been established. In this study, a conceptual design was established to apply RSIM in the actual field, and a scale model experiment was performed to prove it. As a result of the scale model experiment, the applicability of RSIM was confirmed. Finally, numerical analysis using Abaqus/CAE was performed to carry out the detailed design of RSIM (installation conditions such as dimensions). Analysis parameters were embedded depth, thickness, diameter, and material properties of a supporting pile according to the ground type. As a result of the analysis, it was confirmed that the results of RSIM analysis were interacting with all parameters according to the ground conditions. Therefore, it was confirmed that the actual design should be based on ground investigation and economic conditions, not standardized regulations.

## 1. Introduction

The role of soil in relation to public health is important [1,2,3]. Healthy soil can provide nutritious food and clean water for human health, but when a soil contaminant occurs, it is harmful to the environment, plants, animals, and the human body in the short and long term [4]. A report by the UN (Sustainable Development Goal 3) has also designated it as an international political field [5]. The representative factor negatively affecting soil contamination is oil, which reduces the sustainability of the soil due to its strong toxic effect [6]. That is, it not only negatively affects the ecosystem, but also changes the overall environment [7].

Oil is an essential energy source for human development and is a very important product for the international community [8]. Although alternative resources, such as natural gas and energy, are used for environmental reasons, recent data from several countries confirm that the use of oil for power generation has increased across Europe, the Middle East, and Asia [9]. However, soil contamination may occur due to leakage during extraction, development, transportation, processing, and storage of oil, including improper disposal of petroleum-containing waste, abuse of pipelines, and oil storage tanks [10]. Oil is complex compounds containing saturates, aromatics, and heteroatoms. One of the types of aromatics, polycyclic aromatic hydrocarbons (PAHs), is classified as a toxic substance causing human carcinogenic potential, malformations, and mutations [11,12]. This material has lipophilic properties, so it can be absorbed by the human body from the lungs, intestines, and skin, and has a detrimental effect on ecosystems and plants when leaked into soil and groundwater [4,13].

According to references [14] related to oil spills from global tankers, between 1970 and 2018, there were a total of 1702 officially reported oil spills of 7 tons or more, and more spills may have occurred unofficially. The majority (86%) of all spills occurred in marine environments, such as oil tankers. Onshore spills were mostly from pipelines until the early 2000s, but this decreased to 6% of all accidents between 2010 and 2018. Conversely, land-based facilities and tanks, which had less than 10 spills in the decade from 1970, had about 90 spills between 2010 and 2018, accounting for 20% of all spills during that period. In Malaysia, a large amount of oil leakage has been reported due to technical defects in crude oil storage facilities, and concerns have been raised about accidental oil leakage from various onshore facilities [15,16]. In addition to Nigeria’s inland areas, the Niger Delta is a prime example of an oil spill area. In this area, where there is abundant oil, the environment and health of citizens are threatened due to an unknown oil spill problem. Accordingly, petroleum activities are regulated and controlled according to the Environmental Guidelines and Standards for the Petroleum Industry in Nigeria (EGASPIN) [17,18].

Therefore, when an oil spill occurs, the area should be blocked and restored, which has been suggested by various researchers. In the case of the blocking method, a vertical wall is installed around the contaminated area to block the movement of liquid substances, including groundwater. Representative examples include mud wall, grouting wall, sheet-pile wall, deep soil mixing, geomembrane, and lining technology. The permeability of the barrier material needs to be less than 10^−7^ m/s permeability [19,20,21]. After that, remediation methods for contaminant soil are carried out. Remediation methods, such as incineration, soil washing, soil vapor extraction, and biological methods, and thermal decomposition are mainly performed, and variously applied depending on the environment [22,23,24]. However, the preceding methods correspond to post-treatment of contaminant soil, so the identification or detection of oil spill is more important than this. If oil spill is confirmed late, the degree of contamination may become serious, which causes great economic, social, health, and environmental problems.

The smart liner being developed in Korea can check for oil spills. It is an alternative to oil spills in that it is installed in a completely closed state in areas where oil spills are expected to occur [25,26,27]. The smart liner allows the flow of groundwater under normal conditions, but it is a special fiber inserted with a compound that reacts only to oily substances (especially PAHs, such as LNAPL and DNAPL). When oil comes into contact with it, it swells and reduces the permeability coefficient to less than 10^−7^ m/s. When the flow of groundwater is interrupted by blockage of hydraulic behavior, it is detected by an external collecting well, and it can be confirmed that oil spill has occurred in this area. That is, the smart liner is used to prevent and detect the spread of a soil contaminant by oil spill. The smart liner should be installed vertically in the ground, but it is difficult to install because it is in the form of fiber. Unlike the materials of the existing vertical barrier method that block the flow of groundwater, the smart liner does not have rigidity, so it is impossible to insert it into the ground by mechanical pressure. It is necessary to allow the movement of groundwater before an oil spill, so the installation of artificial walls should not be performed, and it is impossible to attach the smart liner to the wall. In addition, the method of directly attaching the smart liner to the vertical excavation surface is difficult due to the irregularity of the excavation surface, and there is a risk of loss after installation.

Therefore, in this study, the vertical construction method of the smart liner in the ground was proposed. This method means that the smart liner in the form of fibers is vertically self-supporting. The production of the smart liner was manufactured as a roll type, and the roll spreading and inserting method (RSIM) was proposed to utilize it. The method is to install a supporting pile after excavation of the surrounding ground in an area where an oil contaminant is predicted, and connect the smart liner in the vertical direction. A detailed explanation is found in Section 3. The reduced model experiment was performed to evaluate the applicability of the proposed method, and the behavior was analyzed using Abaqus/CAE [28], a numerical analysis program, for a detailed design.

## 2. Smart Liner to Prevent Spreading of Oil Spill and Identify Contamination

### 2.1. Behavior and Principle of the Smart Liner

The smart liner is installed vertically in the predicted oil spill area, as shown in Figure 1. The installation depth is up to the aquifer layer, where PAHs, such as LNAPL (light nonaqueous phase liquid) or DNAPL (dense nonaqueous phase liquid), have the potential to move with the flow of groundwater. The installed smart liner enables the movement of groundwater under normal conditions to prevent unpredictable behavior due to groundwater blockage in terms of geotechnical engineering. However, when the PAHs move out of the specific area with the flow of groundwater, the smart liner acts as a vertical barrier through adsorption swelling, and this agglomeration behavior blocks hydraulic behavior, such as the flow of groundwater. At this time, the flow of groundwater is detected from the external collecting well, and if hydraulic behavior is stopped, immediate remediation is performed. Therefore, the use of the smart liner is a kind of preventive method to reduce damage caused by oil spill.

### 2.2. Manufacture Process of the Smart Liner

The basic composition of the smart liner is a sandwich-type mat, as shown in Figure 2. The oil-absorption resin for the reaction of oil is located in the central part, and the top and bottom are combined with nonwoven fabric to form a flexible mat. Here, the oil-absorption resin is composed of a special material that exhibits adsorption swelling and cohesive behavior when oil contaminants leak and contact with the smart liner. The nonwoven fabric plays the role of fixing the oil-absorption resin powder. At this time, the reason for selecting a nonwoven fabric as a material to wrap the oil-absorption resin is that it must have permeability to allow the flow of groundwater under general conditions (when the oil contaminant does not occur).

The smart liner manufacturing process is shown in Figure 3. In the conceptual aspect of Figure 3a, the nonwoven fabric is positioned at the bottom, and the oil-absorption resin is evenly positioned on the bottom of the nonwoven fabric through the resin injection part. After that, the upper nonwoven fabric is placed on the oil-absorption resin, and three layers are attached and integrated through a needle punching machine. Needle punching not only fixes the three layers, but also distributes the oil-absorption resin evenly so that it can exist within a certain range of the nonwoven fabric. Design drawings and a picture of actual equipment are shown in Figure 3b, and the maximum production width of the smart liner is 3.0 m. The finally manufactured smart liner is stored and transported in a roll type, as shown in Figure 3c for various conveniences.

## 3. Proposal of Construction Methods of the Smart Liner

### 3.1. Definition of the Roll Spreading and Inserting Method (RSIM)

RSIM is a construction method in which the smart liner produced in a roll type is completely straightened and inserted into the supporting pile using construction devices, such as a backhoe or crane. The supporting pile is a kind of pile structure used for vertical location of the smart liner, and is penetrated into the excavated ground. Both ends of the unfolded smart liner are attached to the connecting stick and inserted into the supporting pile. Finally, each supporting pile is connected by a smart liner and blocks hydraulic behavior in case of oil spill. A schematic diagram of RSIM is shown in Figure 4.

In detail, the shape of the supporting pile and the connection between the supporting pile and the smart liner are shown in Figure 5a. The supporting pile has grooves in four directions, and inside has an ‘X’ shape for stability. The connecting stick attached to the smart liner is inserted into this groove, and the smart liner is connected to the groove. The reason for setting the groove is that there may not be a straight section including the right-angle part during the connection process. In the construction process of the smart liner, the supporting pile is penetrated in consideration of the length of the smart liner, but it is impossible to calculate the perfect distance, so it is constructed with an extra length. Therefore, the tension of the smart liner cannot be maintained, as shown in Figure 5b, which may cause problems in the process of blocking oil spill. To solve this, the supporting pile must be rotated after the insertion of the smart liner, which contributes to the tension of the smart liner.

### 3.2. Process of the Roll Spreading and Inserting Method (RSIM)

The construction sequence of RSIM is as follows: (1) excavation, (2) penetration of the supporting pile, (3) connection within the support after unfolding the smart liner, (4) maintaining the tension of the smart liner with the rotation of the supporting pile, (5) grouting for flattening the excavation lower surface, (6) backfill. At the time, the excavation method and process are not considered in this manuscript.

#### 3.2.1. Installation of the Supporting Pile and Smart Liner

The supporting pile is penetrated into the ground where the trench excavation has been completed. Then, the connecting stick attached to both ends of the fully unfolded smart liner is inserted into the supporting pile, as shown in Figure 6.

#### 3.2.2. Maintaining the Tension of the Smart Liner

A smart liner without tension can cause problems in the process of reacting to oil contamination, so the smart liner must be fully flattened. For this, the supporting pile is rotated through the rotating equipment, and tension is secured by adjusting the length of the smart liner, as shown on the right side of Figure 7. The inside of the supporting pile where the smart liner is inserted is filled with a fluid filling material or foamed urethane. This is to completely fix the smart liner and supporting file and to prevent the leakage of groundwater or oil spill that may occur in this part.

#### 3.2.3. Backfill

The RSIM proposed in this study was preceded by trench excavation. However, in the trench excavation process, the lower surface has an irregular shape. If the excavation lower surface is not flattened, it may cause a problem, as shown in Figure 8. The separation distance between the excavation lower surface and the smart liner exists, and it has a possibility that oil contaminants may leak through the gap. Therefore, the excavation lower surface must be made flat by injecting a fluid filling material, such as mortar or bentonite cement. At this time, the leveling of the excavation surface may be performed regardless of before or after construction. When leveling is completed, backfill is performed on the side of the excavation, and finally, compaction is carried out to complete the construction.

## 4. Verification of the Applicability of RSIM through the Reduced Model

### 4.1. Ground Composition

The chamber to be used in the experiment was 1.0 m in width and length and 0.6 m in height, the frame was made of steel, and the front was made of transparent acrylic such as Figure 9. The ground was composed of standard sand and compacted in 5 layers with sand (the degree of compaction was 90%).

### 4.2. Reduced Model of the Supporting Pile

The design dimensions of the supporting pile for the applicability evaluation of RSIM is shown in Figure 10a, and the actual production is shown in Figure 10b. The reduced scale was 1/30, and the supporting pile was manufactured through steel processing. The combination of the smart liner and the manufactured supporting pile is shown in Figure 10c. The smart liner was simulated with only one nonwoven fabric. The penetration depth in the ground of the supporting pile was set to 50 mm. Initially, the reduced model was attempted to further reduce the diameter while maintaining the height. It was difficult to simulate a smaller cross section during the steel processing using a machine. This experiment was simply to verify the suitability of the method proposed in this study. Therefore, the shape of the supporting pile was important, and the dimensions and material were not important at this point. Specific design dimensions will be carried out in Section 5.

### 4.3. Case

The experiment was to confirm whether RSIM’s supporting pile can be applied to various construction site areas and the smart liner can maintain tension after construction. In terms of geotechnical engineering, bearing capacity and displacement were not measured. The experimental case for this is shown in Figure 11. The construction shape of RSIM is a square and multisection area, and the installation position of the supporting pile is at each corner. Additionally, if the distance between the supporting piles is far, the tension of the nonwoven fabric may not be maintained. Therefore, in the rectangular and multisection area, one additional supporting pile was placed between the supporting piles at the corner.

### 4.4. Results

In the experimental process, excavation is carried out by the construction area first, and then the supporting piles and nonwoven fabric are installed. At this time, if it is impossible to maintain the tension of the nonwoven fabric, the supporting piles must be rotated. The experimental results are shown in Figure 12 and Figure 13. The nonwoven fabric could not maintain tension in the connection process. It was confirmed that when rotation was applied to the supporting pile, the nonwoven fabric was fully unfolded. As a result of the experiment, the geometrical suitability of RSIM for smart liner installation was confirmed. In addition, it was shown that the rotation radius of the whole supporting pile decreased when an additional supporting pile was installed between the supporting piles at all corners.

### 4.5. Manufacture of Construction Models Made in a 1/3 Scale

Finally, a 1/3 scale model was additionally manufactured to confirm the applicability of RSIM. The purpose of the manufactured model was to verify the actual ground penetration and connecting process, and the complete closure of the construction ground was not considered. As a result of the verification, as shown in Figure 14, it was confirmed that there were no problems caused by the penetration and rotation of the supporting pile in the real environment, and that the tension of the nonwoven fabric simulated by the smart liner material was maintained. At this time, the supporting pile was made of aluminum, and the cross section was different from the existing design during the manufacturing process. However, there was no significant difference in the connecting process.

## 5. Numerical Analysis for a Detailed Design

The applicability of RSIM proposed in this study was confirmed through the results in Section 4. This is only for the verification of the shape and construction method. Therefore, it is necessary to determine the specific design dimensions and materials. For this purpose, in this study, numerical analysis is performed, and the analysis program is Abaqus/CAE [28].

### 5.1. Material Properties

Table 1 shows the material properties of the supporting pile used for numerical analysis. For the material of RSIM, steel, aluminum, and HDPE (high-density polyethylene) were considered. The ground was considered weathered soil and rock, which are relatively hard ground because the lower surface of the excavation to be constructed is set below the aquifer.

### 5.2. Analysis Cases

The analysis cases of RSIM are shown in Figure 15 and Table 2, and there are a total of 54 cases. The analysis parameters are material properties, thickness, embedded depth of the supporting pile, and ground. The diameter of the supporting pile is fixed at 0.3 m. The height of the smart liner is fixed at 3.0 m (maximum height of the smart liner that can be manufactured at this time), and the total length of the support is equal to the penetration depth plus 3.0 m.

### 5.3. Modeling of Numerical Analysis

#### 5.3.1. Modeling

The supporting pile is shown in Figure 15 and Table 2, and the ground is a cylindrical structure with a diameter of 10.0 m and a height of 5.0 m (Figure 16). The composition of the mesh is a three-dimensional element, the element type is ‘3D Stress’, and C3D8R (an 8-node linear brick, reduced integration, hourglass control) is set in consideration of the deformation factors of the ground and supporting pile.

#### 5.3.2. Boundary Condition

As a boundary condition, the movement of the side of the ground is fixed by assuming that there is infinite ground as a roller condition. The lower surface of the ground is a hinge condition, and if not fixed, the entire modeling structure descends downward. At this time, the modeling is a three-dimensional element composed of x-, y-, and z-coordinates. In this study, the x- and y-axes are set as planes, and the z-axis is set as height. Accordingly, in numerical analysis, the side boundary condition can be defined as fixing the horizontal plane of ‘x = 0, y = 0’, and the bottom boundary condition as fixing the vertical plane of ‘z = 0’.

#### 5.3.3. Gravity Condition

In the numerical analysis, the modeled ground and supporting pile represent a zero-gravity state, and in general, ground structures should consider the load by gravity, so it was applied to the entire model. The actual ground and supporting piles are under gravity, and stress by gravity exists, but displacement does not occur. However, the analysis program recognizes gravity as an additional load when it is applied. Therefore, stress is generated, as well as additional displacement of the modeling or deformation of the mesh. Therefore, it is necessary to restore the initial state that can reverse the displacement or deformation caused by gravity. This analysis program (Abaqus/CAE) defines it as geostatic, and this study reflects it in the program.

#### 5.3.4. Numerical Simulation for the Tension of the Smart Liner

After the application to the initial state by gravity was completed, displacement control was performed, as shown in Figure 17, to simulate the tension of the smart liner. The connecting pipe and smart liner were not simulated for the convenience of numerical analysis, and only the one-way load with the rotation of the supporting pile was set. The location of displacement control was the groove surface of the supporting pile where the smart liner was pulled. At this time, the displacement control was 0.3 m, which was 10% of the diameter of pile head that the lateral capacity of a general pile was measured.

### 5.4. Results of Numerical Analysis

As a result of the numerical analysis, the load and displacement are shown in Figure 18, Figure 19, Figure 20, Figure 21, Figure 22 and Figure 23. At this time, the load (t/m) is the ratio of the reaction force with the displacement control and the height of the smart liner, 3.0 m. This means the allowable load per unit length. That is, the tensile strength of the smart liner must be less than the allowable load. If the tensile strength is greater than the allowable load, it may be impossible to maintain the tension of the smart liner. Additionally, the overturning or failure of the supporting pile may occur during the rotation process to maintain the tension.

For the data in Figure 18, Figure 19, Figure 20, Figure 21, Figure 22 and Figure 23, the allowable load is shown in Figure 24 and Table 3. Here, the allowable load was the load per unit length when the displacement was 0.3 m. Depending on the ground conditions, the allowable load showed different tendencies. In general, the allowable load had high value when the embedded depth, the elastic modulus of the material, and the thickness of the supporting pile cross-section were large. However, in some conditions, the allowable load decreased as the embedded depth increased.

In the weathered soil in Figure 24a, steel showed a clear difference in allowable load with the thickness, and aluminum had a similar value when the allowable load was 0.10D or more. For HDPE, the allowable load increased according to the embedded depth, and it was confirmed that the effect of the thickness was insignificant.

In the weathered rock in Figure 24b, there was a clear difference in all material properties according to the thickness, unlike the weathered soil. At this time, the effect according to the embedded depth was irregular, but in the case of HDPE, it was confirmed that the embedded depth was hardly affected.

## 6. Conclusions

This study proposed a vertical construction method for a smart liner installed to block soil contaminants by oil spill. For this, applicability was evaluated by a conceptual design and the production of a reduced model. Finally, numerical analysis was performed for a detailed design, and the conclusions are as follows:(1)A smart liner cannot use the existing method for a vertical barrier in the ground; therefore, a connection method using a supporting pile and a connecting stick is proposed. In terms of idea, this is advantageous, and the rotation of the supporting pile for the tension of the smart liner and the groove for the connecting process were considered.(2)The suitability of the conceptual design process was confirmed through experiments using a reduced model. The proposed method can be applied not only to a square area but also to multisections. The extra length of the smart liner that can occur during the construction process can be controlled through the rotation of the supporting pile, which is also suitable for maintaining tension. As a result of verifying the optimal construction length (distance between supporting piles) for this purpose, it was observed that there was no significant correlation in the current experimental stage, but variables may occur for actual construction.(3)From the numerical analysis results, an increase in the allowable load was expected as the embedded depth increased, but trends varied. This is because various stresses occurred on the lower surface of the supporting pile, depending on the material properties and depth in the process of lateral displacement. In other words, the embedded depth is the most important factor in the design of supporting piles, and the analysis with various embedded depths must be preceded before actual construction.(4)In the material properties, steel was affected by all analysis variables, and the allowable load was also large. Aluminum is also similar to steel, but it is necessary to secure a certain thickness in weathered soil. HDPE showed the most consistent tendency, but it has a disadvantage that the allowable load is very low. Accordingly, HDPE, which can be used by the public, is suitable when a large load is not expected. Conversely, if a large load is expected, numerical analysis must be performed, considering the transportation cost and manufacturing cost of steel and aluminum. Appropriate design parameters and materials should be selectively used under all conditions.(5)In future research, RSIM will be planned to be constructed in the actual field, and an actual design will be performed. The actual design aims to provide materials and dimensions optimized for the ground conditions of the construction site. Optimization here means considering economic feasibility and constructability. In addition, in this study, the groundwater blocking of the smart liner due to oil contamination was not verified, but this will be verified in future studies.

## Figures and Tables

**Figure 1 ijerph-20-00940-f001:**
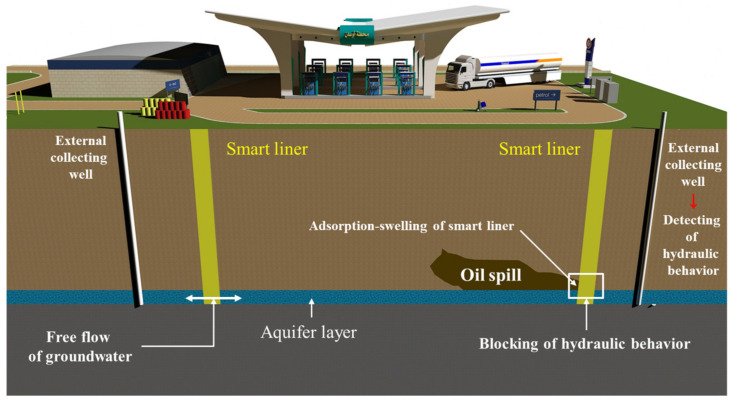
Installation of the smart liner and behavior in the ground with oil spill.

**Figure 2 ijerph-20-00940-f002:**
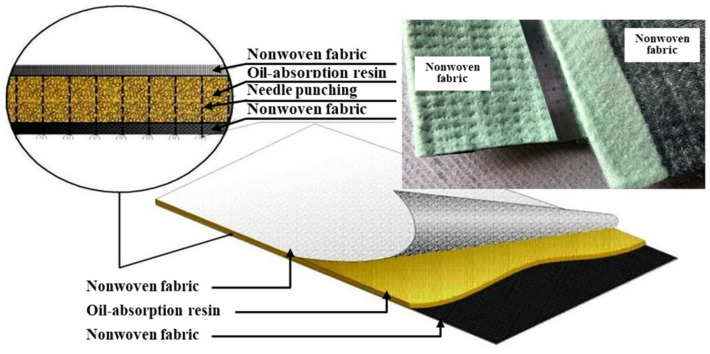
Composition of the smart liner.

**Figure 3 ijerph-20-00940-f003:**
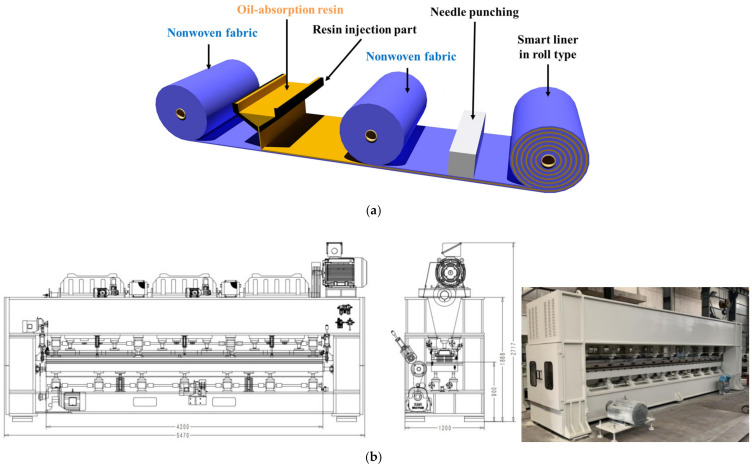
Manufacture of the smart liner: (**a**) concept diagram, (**b**) design drawings, actual picture of the manufacture equipment, and (**c**) sample of the smart liner produced in a roll type.

**Figure 4 ijerph-20-00940-f004:**
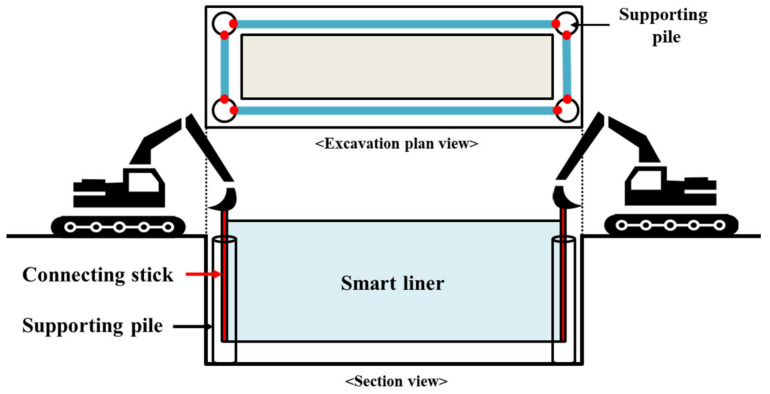
Schematic diagram of RSIM.

**Figure 5 ijerph-20-00940-f005:**
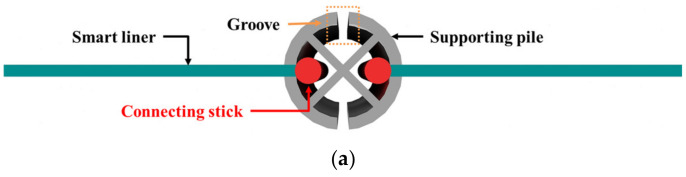
Detailed concept design of RSIM: (**a**) shape of the supporting pile and connection between the supporting pile and the smart liner and (**b**) maintenance method for the tension of the smart liner.

**Figure 6 ijerph-20-00940-f006:**
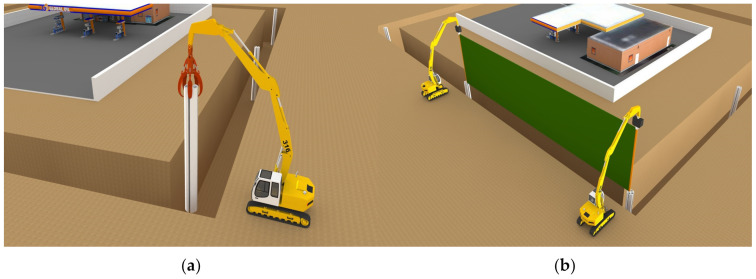
Installation of (**a**) the supporting pile and (**b**) the smart liner.

**Figure 7 ijerph-20-00940-f007:**
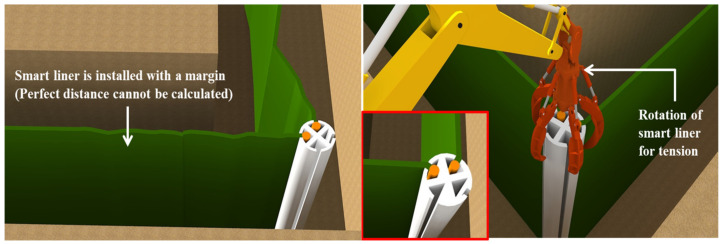
Maintenance method for the tension of the smart liner.

**Figure 8 ijerph-20-00940-f008:**
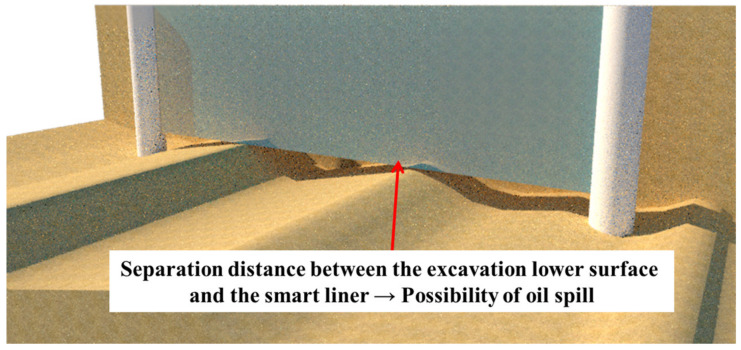
Problem when the leveling of the excavation lower surface is not performed.

**Figure 9 ijerph-20-00940-f009:**
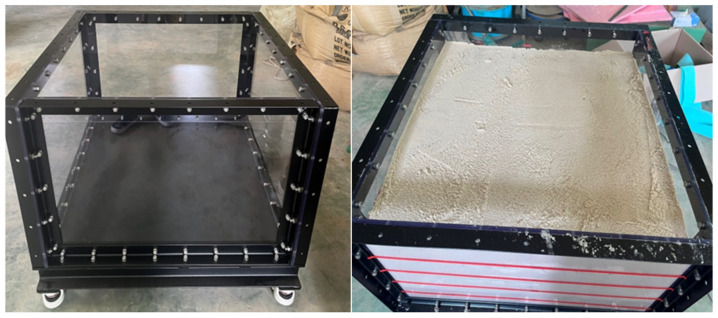
Ground composition.

**Figure 10 ijerph-20-00940-f010:**
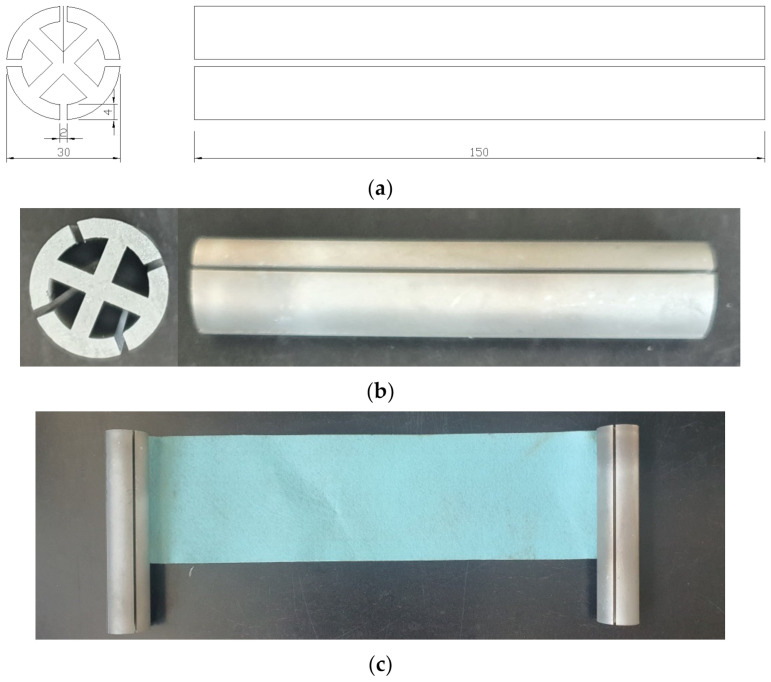
Reduced model of RSIM: (**a**) design dimensions (unit: mm), (**b**) actual production, and (**c**) combination of the supporting pile and the smart liner.

**Figure 11 ijerph-20-00940-f011:**
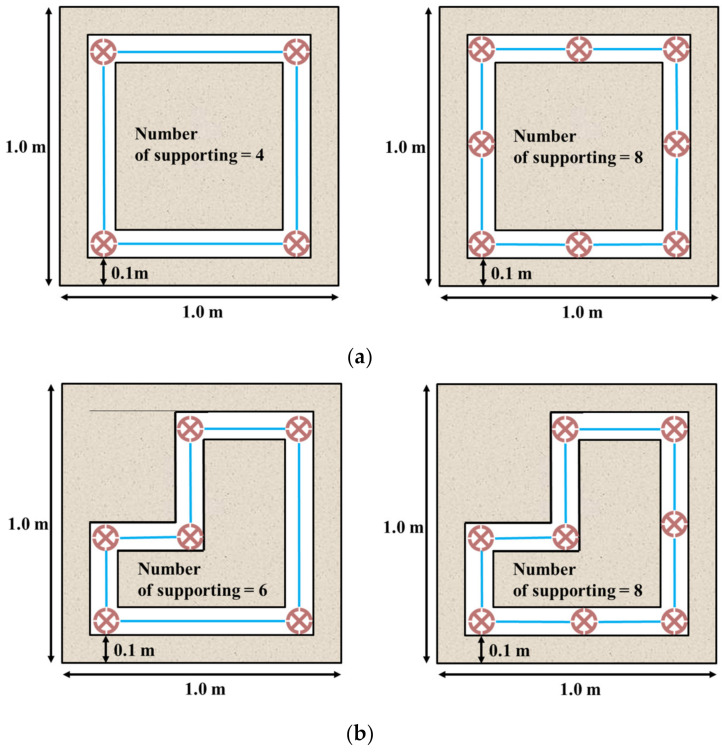
Cases of experiments for RSIM: (**a**) square and (**b**) multisection area.

**Figure 12 ijerph-20-00940-f012:**
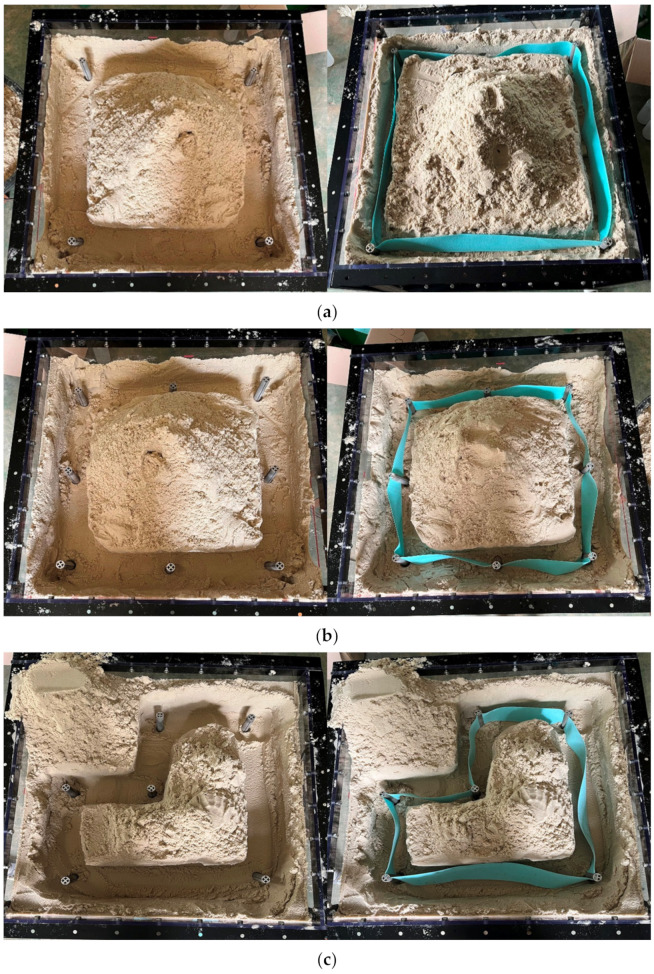
Results of experiments for RSIM: square using (**a**) 4-supporting, (**b**) 8-supporting, and (**c**) multisection area.

**Figure 13 ijerph-20-00940-f013:**
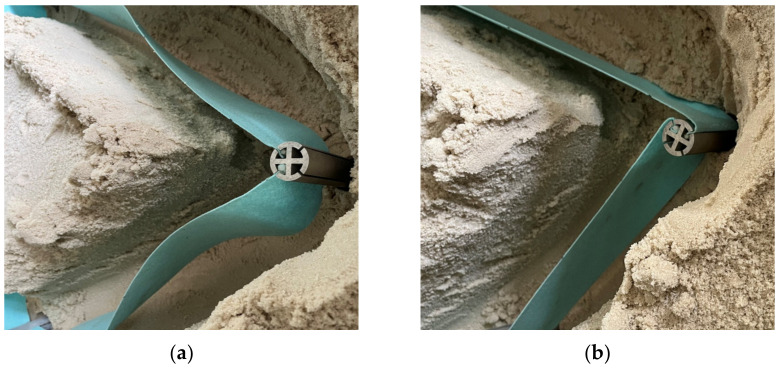
Tension of the nonwoven fabric (**a**) before and (**b**) after rotation of the supporting pile.

**Figure 14 ijerph-20-00940-f014:**
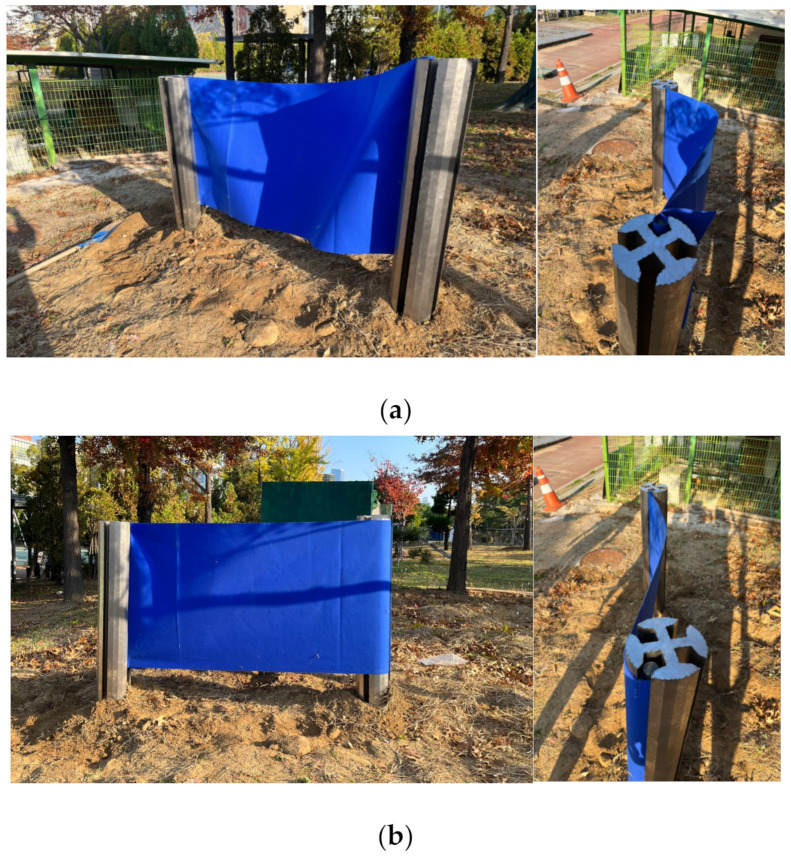
Tension of the nonwoven fabric (**a**) before and (**b**) after the rotation of the supporting pile.

**Figure 15 ijerph-20-00940-f015:**
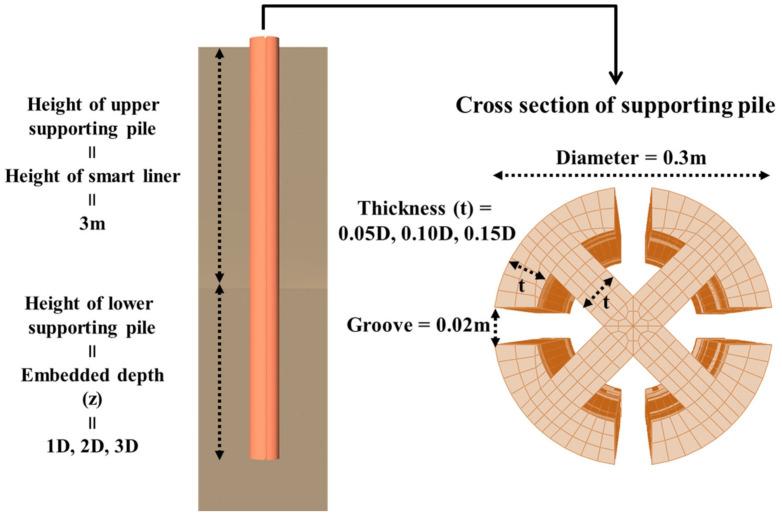
Analysis parameters of RSIM.

**Figure 16 ijerph-20-00940-f016:**
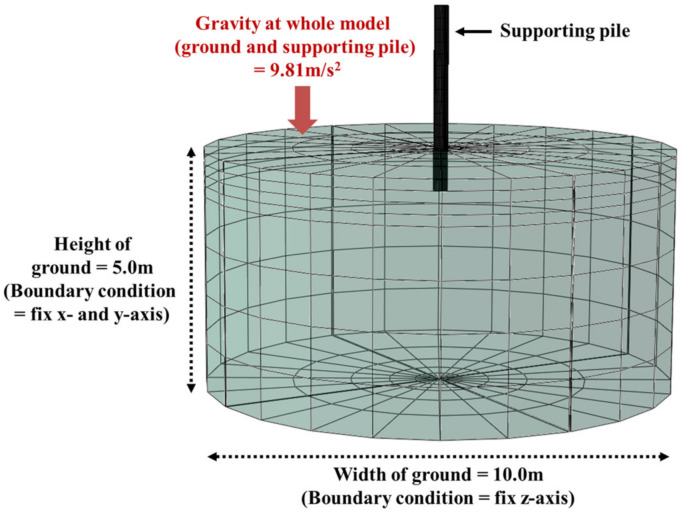
Modeling of RSIM.

**Figure 17 ijerph-20-00940-f017:**
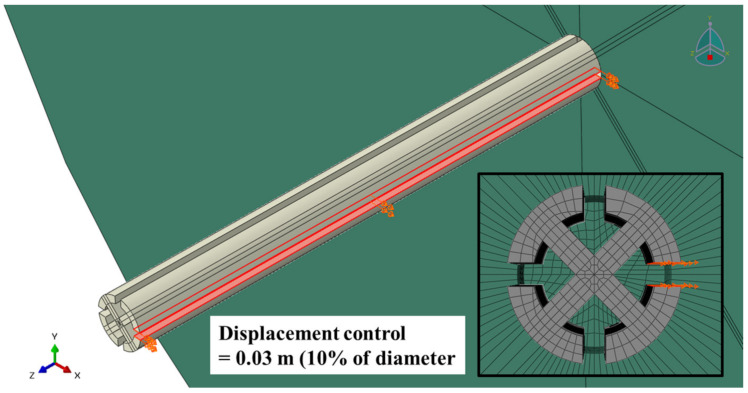
Tension simulation of the smart liner.

**Figure 18 ijerph-20-00940-f018:**
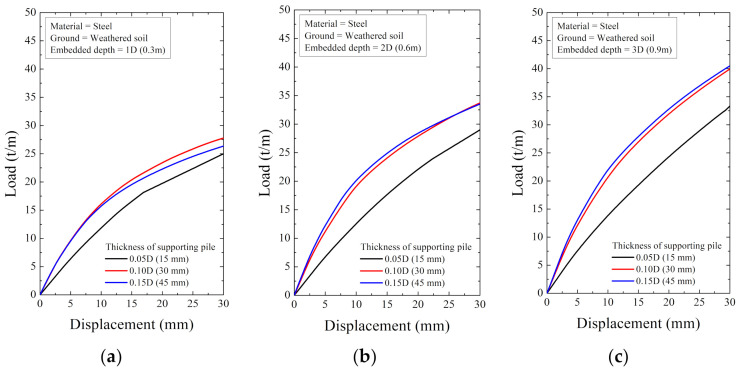
The displacement–load curve when the supporting material is steel in weathered soil at embedded depth: (**a**) 1D (0.3 m), (**b**) 2D (0.6 m), and (**c**) 3D (0.9 m).

**Figure 19 ijerph-20-00940-f019:**
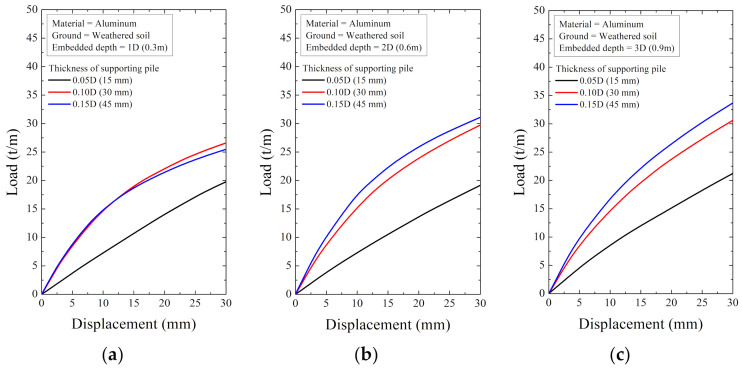
The displacement–load curve when supporting material is aluminum in weathered soil at embedded depth: (**a**) 1D (0.3 m), (**b**) 2D (0.6 m), and (**c**) 3D (0.9 m).

**Figure 20 ijerph-20-00940-f020:**
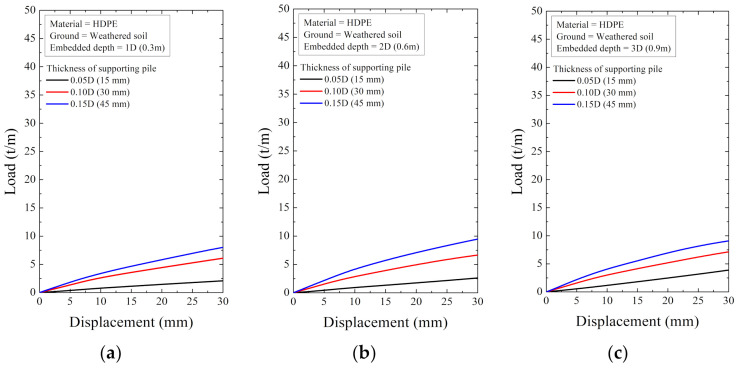
The displacement–load curve when supporting material is HDPE in weathered soil at embedded depth: (**a**) 1D (0.3 m), (**b**) 2D (0.6 m), and (**c**) 3D (0.9 m).

**Figure 21 ijerph-20-00940-f021:**
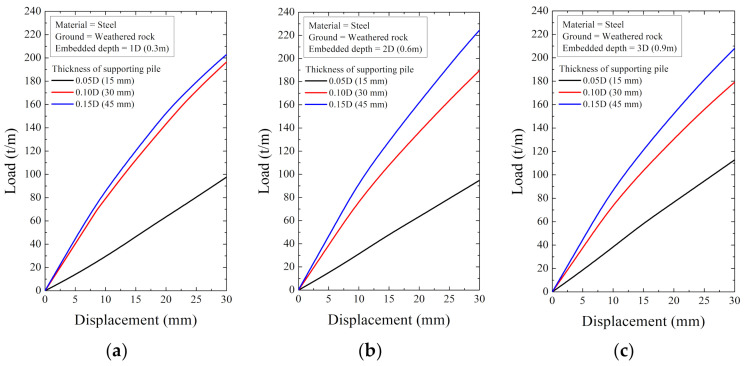
The displacement–load curve when supporting material is steel in weathered rock at embedded depth: (**a**) 1D (0.3 m), (**b**) 2D (0.6 m), and (**c**) 3D (0.9 m).

**Figure 22 ijerph-20-00940-f022:**
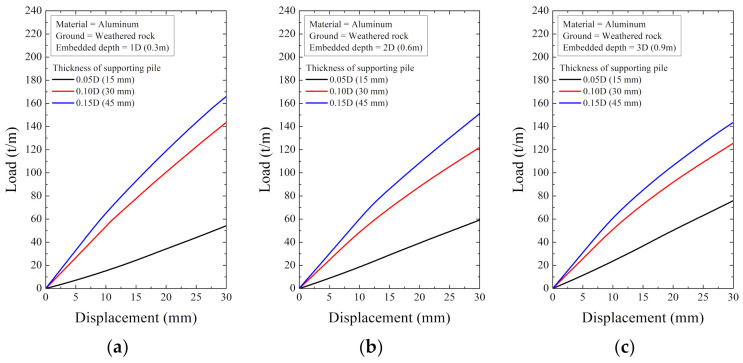
The displacement–load curve when supporting material is aluminum in weathered rock at embedded depth: (**a**) 1D (0.3 m), (**b**) 2D (0.6 m), and (**c**) 3D (0.9 m).

**Figure 23 ijerph-20-00940-f023:**
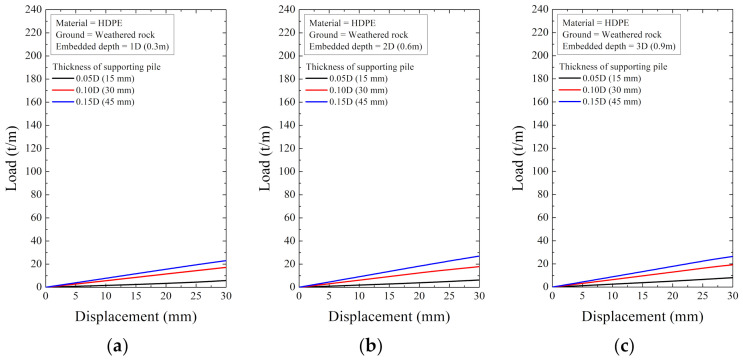
The displacement–load curve when supporting material is HDPE in weathered rock at embedded depth: (**a**) 1D (0.3 m), (**b**) 2D (0.6 m), and (**c**) 3D (0.9 m).

**Figure 24 ijerph-20-00940-f024:**
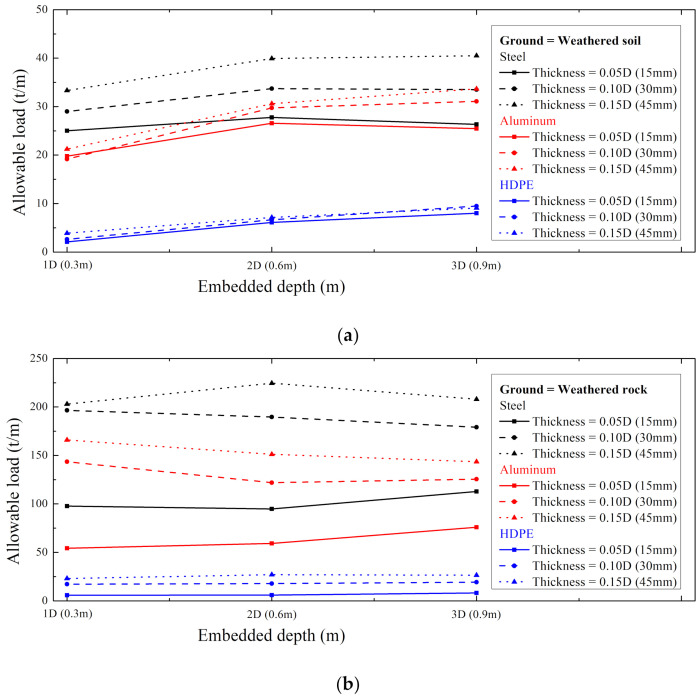
The maximum load with embedded depth in (**a**) weathered soil and (**b**) rock.

**Table 1 ijerph-20-00940-t001:** Material properties of the ground and supporting pile.

Material Properties	Weathered Soil	Weathered Rock	Steel	Aluminum	HDPE
Model	Mohr–Coulomb			Elastic	
Density, γ (kN/m^3^)	19.0	21.0	78.0	27.0	9.5
Elastic modulus, E (MPa)	100	500	210,000	70,000	1000
Poisson’s ratio, υ	0.3	0.3	0.3	0.33	0.42
Internal friction angle (°)	33	35	-	-	-
Cohesion, c (kPa)	180	2500	-	-	-
References	[29]	[30]	[31]	[31]	[32]

**Table 2 ijerph-20-00940-t002:** Analysis cases of RSIM.

Ground	Supporting Pile			
	Material	Diameter, D	Thickness, t	Embedded Depth, z
Weathered soil and rock	Steel	0.3 m	0.05D	1D
	Aluminum		0.10D	2D
	HDPE		0.15D	3D

**Table 3 ijerph-20-00940-t003:** Allowable load of RSIM.

Ground	Supporting Pile		Load (t/m)		
	Embedded Depth	Thickness	Steel	Aluminum	HDPE
Weathered soil	1D (0.3 m)	0.05D (15 mm)	25.04	19.78	2.09
		0.10D (30 mm)	27.79	26.59	6.10
		0.15D (45 mm)	26.35	25.48	8.02
	2D (0.6 m)	0.05D (15 mm)	29.00	19.16	2.62
		0.10D (30 mm)	33.72	29.74	6.66
		0.15D (45 mm)	33.52	31.10	9.48
	03D (0.9 m)	0.05D (15 mm)	33.33	21.23	3.89
		0.10D (30 mm)	39.94	30.61	7.15
		0.15D (45 mm)	40.50	33.67	9.09
Weathered rock	1D (0.3 m)	0.05D (15 mm)	97.64	54.29	5.84
		0.10D (30 mm)	196.51	143.56	17.21
		0.15D (45 mm)	202.94	166.01	23.05
	2D (0.6 m)	0.05D (15 mm)	94.80	59.25	5.89
		0.10D (30 mm)	189.63	121.97	17.88
		0.15D (45 mm)	224.54	151.20	27.00
	03D (0.9 m)	0.05D (15 mm)	112.80	75.92	8.14
		0.10D (30 mm)	179.18	125.57	19.24
		0.15D (45 mm)	208.03	143.58	26.51

## Data Availability

Not applicable.

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
