# Peer review of "Proposal of Construction Method of Smart Liner to Block and Detect Spreading of Soil Contaminants by Oil Spill"

_ijerph, 2023, doi:10.3390/ijerph20020940_

Round 1

Reviewer 1 Report

Dear Authors,

I have reviewed the paper entitled "Proposal of Construction Method of Smart Liner to Block and Detect Spreading of Soil Contaminants by Oil Spill". The papers' methods and quality of presentation is very good. I would like to propose few things to Authors, that I think will improve the overall ratings of the paper. 

1. Abstract is very laconic. Please, add few sentences about methodology, problem statements and quantitive results.

2. The Introduction is not written in a professional matter. I think that the Authors should include more references, especially those which underline the Authors contribution.

3. There are many possibilities for oil spills to occur (f.e. tank catastrophies). From the Abstract and the Introduction I dont know whats' exact case study Authors would like to propose.

4. Conclusions are underlined and its very good. However I would like to know from the paper what are the next steps of research that the Authors would like to conduct. Please, write few sentences about future work.

Kind regards/

Author Response

Thanks to the reviewer's opinion, the corrected file is attached as follows.

Reviewer 2 Report

This design does not show the source of the oil spill and is not suitable for spills from pipelines. The solution will be impractical for pipeline spills as the time required to set up will defeat the purpose for setting up the containment.  The research doesn't show how the entire width of the underground water way will be contained and the role of the well was not adequately discussed. It also does not show how the oil spill occurrence is predicted. The references are also grossly inadequate . The use case appears to be a gas station with underground tank. If this is the case, it should be clearly presented as such both in the title and in the body of the work as a general reference to oil spills will render the solution impracticable.

Author Response

(The authors gave the same response as above.)

Round 2

Reviewer 2 Report

The addition of the scope of operation of the liner, has improved the value of the paper.